# Combined models of violent conflict and natural hazards improve predictions of household mobility in Bangladesh
Maxine Leis [1] ✉ & Kristina Petrova [2] ✉

In 2023, the United Nations High Commissioner for Refugees reported over 110 million displaced individuals globally, many in regions facing extreme weather and violence. Here we examine how these crises interact to shape household mobility in Bangladesh. Using data linking local conflict events, natural hazards, and household characteristics from 2011 to 2018, we apply machine learning models to capture complex, non-linear relationships between these risks. We find that combining conflict and hazard information improves predictions of household mobility. While exposure to violence or disasters increases mobility, households with remittances are more likely to move, whereas those with loans often remain. Interactions, such as between one-sided violence and landslides, further amplify movement, highlighting the importance of understanding how multiple stressors jointly influence household decisions.

Extreme weather events and natural hazards increasingly strike many parts of the world and often impact countries already destabilized by political insecurity. These crises do not occur in isolation. Rather, they are likely to interact in complex ways, shaping household decisions under highly uncertain conditions. Bangladesh exemplifies this reality. For decades, the country has been exposed to recurrent natural hazard-related disasters, serving as a testament to both resilience and vulnerability amidst these challenges. Despite significant progress in climate adaptation and disaster preparedness, Bangladesh's newly established interim regime continues to contend with political insecurity, underscoring the compound challenges the nation and its population face[1]. This convergence of crises in Bangladesh may foreshadow the future of other climate- and violence-affected regions. As climate change intensifies, many countries, especially those with fragile political systems and limited governance capacity, may experience similar compounding insecurities, further shaping mobility decisions in unpredictable ways.

Understanding how households navigate these overlapping risks is therefore critical for anticipating migration patterns and informing policy responses. However, scientific knowledge on the specific interaction between natural hazards and different forms of violent conflict remains largely underexplored. A substantial body of research explores the influence of natural hazards on conflict risk[2] and the ways in which disasters exacerbate the vulnerabilities that predispose societies to conflict[3]. While previous studies have examined the causal relationship between climate, armed conflict, and asylum seeking, as well as the broader climate-conflict-refugee nexus[4,5], such integrated approaches to

studying political and environmental factors remain rare. Our work posits that while the individual impacts of natural hazards and political crises on human mobility are well known, their combined effects might create new and complex challenges[6]. Despite the increasing visibility of disaster impacts, particularly in politically unstable regions, there still exists an urgent need for understanding how these intertwined factors collectively influence migration trends[7,8]. The complexity of these crises, particularly in fragile settings, is what motivates our response to this research gap.

Our study focuses on first examining both the separate and combined impacts of different forms of violent conflict and natural hazards on the predicted probability of household members migrating by analyzing data from over 5500 households in Bangladesh for the period between 2011 and 2018. In doing so, we reveal how different types of events uniquely influence these migration patterns and emphasize the importance of considering the simultaneous occurrence of these calamities in both space and time. This approach underscores the poorly understood joint risk these factors collectively impose on communities, which is overlooked when studied in isolation. Second, while conflict and disaster exposure may be linked to a greater propensity for migration, these stressors can also limit the ability to do so, depending on household characteristics. Focusing on uneven capabilities for mobility is crucial in studying distress movements, as it helps us understand who possesses the means to become mobile during crises[9]. To account for such dualities, we analyze how migration likelihood shaped by these events varies across different household characteristics. In line with findings from prior research[10,11], we challenge the idea of a uniform response

[1]Department of Peace and Conflict Research, Uppsala University, Uppsala, Sweden. [2]Potsdam Institute for Climate Impact Research (PIK), Member of the Leibniz Association, P.O. Box 60 12 03, 14412 Potsdam, Germany. ✉e-mail: maxine.leis@pcr.uu.se; kristina.petrova@uni-mannheim.de

to climate and violent conflict events on migration, demonstrating that their effects differ even within the same communities.

To achieve our objectives, we employ machine learning algorithms within an out-of-sample prediction framework for our analysis. Machine learning techniques, although rarely applied in the field of distress mobility research, offer unique advantages. An exception in the existing literature applies a random forest algorithm to identify key predictors of migration from a broad set of variables based on variable importance scores[12]. Unlike traditional regression models, machine learning can uncover complex and previously unidentified patterns, capturing non-linear relationships and interactions that may not be immediately apparent. Although these methods cannot isolate causal relationships, they are particularly useful for identifying potential associations between variables that traditional approaches might overlook[13–15]. Moreover, regression-based models often struggle to account for interdependencies arising from endogenous processes. In contrast, a cross-validation framework allows us to evaluate whether specific predictors contribute meaningfully to migration likelihood while enhancing predictive accuracy compared to a benchmark model[16]. Finally, by evaluating the model's predictive accuracy on unseen data, we mitigate the risk of overfitting, ensuring that our findings generalize beyond the training data. While machine learning models can be complex and less transparent, we apply interpretability techniques such as SHAP (SHapley Additive exPlanations) summary plots, which quantify each variable's contribution to the model's predictions.

In our approach, model specification is essential[17]. For feature selection, we build on a theoretical framework that sees migration as driven by "abilities and aspirations": individuals or households aspire to improve their lives by moving, but their ability to do so depends on their resources[18]. This is not to suggest that aspirations to migrate primarily drive people's decisions to flee when confronted with sudden disasters or political violence. Rather, it acknowledges that a significant portion of the population in areas of violence or severe drought may choose to stay despite the inherent insecurity[19,20]. In this context, levels of aspiration can help explain why migration might be a voluntary choice for some, while for others, it becomes an unwanted necessity. We define aspiration as the "conviction that migration is preferable to staying" and ability as the "capacity to act on this wish within specific constraints" p. 946[21]. Our theoretical expectations suggest that individual and household traits shape both the aspiration and ability to migrate. Although our study does not directly measure these aspirations and abilities, we propose that certain characteristics effectively embody both. Indicators such as education, loans, and remittances capture both the motivation and networks to seek opportunities elsewhere, as well as the financial means to realize such plans. These aspirations and abilities are influenced not only by individual and household traits but also by the presence of calamities. When extreme events and violent conflict occur, we anticipate that these events further shape people's decision-making processes. In the context of violent conflict, civilians with pre-existing migration aspirations and an advantaged status are more likely to have the opportunity to migrate safely and act on those motivations[22]. On the other hand, armed conflict often constrains household livelihoods by destabilizing income and increasing poverty[23], which, while heightening the desire to migrate, also reduces the ability to do so. Similarly, natural hazards often cause impacts such as casualties, psychological effects, and economic damages, leading to loss of income and assets[24]. These consequences are likely to increase the desire to migrate[25], but may also diminish the capability to do so. Given these complexities, we remain agnostic about the specific direction of impact on migratory decisions but aim to investigate how migration decisions differ when households are exposed to only one type of calamity versus experiencing a compound effect.

Our findings reveal three key insights. First, models that account for both violent conflict and natural hazards outperform simpler ones, suggesting that combining these factors enhances the predictive accuracy of household-level mobility, an approach not previously tested. Second, while violence or climate disasters are important predictors, they do not unilaterally determine migration outcomes as households with resources like remittances show a greater likelihood of migration, while those burdened by loans may stay. Finally, interactions between violence, disasters, and household characteristics underscore how dual threats, such as the combination of one-sided violence and landslides in Bangladesh, can amplify migration, emphasizing the need to account for compounding risks to refine our understanding of migration predictors.

## Results
### Analytical approach
As the adverse impacts of climate change intensify, increased exposure to natural hazards has become a pressing reality. Bangladesh serves as a particularly relevant case study due to its geographic location and topography, which make it highly vulnerable to extreme weather events such as flooding and severe riverbank erosion[26]. At the same time, the country's political landscape is marked by significant insecurity and ongoing electoral violence, in the wake of the sudden fall of its increasingly authoritarian government[1,27]. Given the crucial role of both internal and international migration in Bangladesh's economic development, including the establishment of migration corridors as adaptive responses to distress, examining Bangladesh offers a unique opportunity to investigate the relationship between the co-occurrence of natural hazards and violence and migration.

To overcome the challenge of accessing detailed longitudinal data on sub-national internal mobility[28,29], we rely on the Bangladesh Integrated Household Survey (BIHS). The BIHS data encompass three survey waves conducted in 2011/2012, 2015, and 2018/2019, and are representative of rural households across Bangladesh at the division level[30–32]. In terms of operationalization of our target variable, households in the BIHS were asked about each member's migration status over the past 5 years, specifying if and when they had migrated (for more than 6 months). We transformed these responses into a panel dataset covering 2011–2018, creating a binary dependent variable to indicate migration status per month, where 0 denotes no migration and 1 indicates migration. The data were then aggregated at the household year level to address recall bias. Table S4 in the SI reports the number and percentage of households experiencing migration events by year. Our analysis reveals that around 30% of the households experienced at least one migration event between 2011 and 2018, with multiple members migrating in about 6% of the households. At the household-year level, however, only about 4.5% of observations record a migration event, underscoring both the rarity of the outcome and the difficulty of predicting household-level migration. Table S5 summarizes the distribution of surveyed households across districts, showing both absolute counts and relative percentages.

To merge the data with information on natural hazards and different forms of violent conflict, households are geo-referenced to the second-order administrative divisions. This method assesses whether households are located in districts affected by specific disasters or types of conflict. Although it might seem challenging to assign households to a district and assume uniform experiences of violence, our approach is designed to capture both direct and more distant exposures to violence, including perceptions of threat and fear. This methodology is consistent with common practices in the field and addresses limitations due to the absence of violent conflict and disaster exposure data in the survey questionnaire. All data and code are available at the project's GitHub repository[33].

In our empirical analysis, we specify four thematic models, as depicted in Fig. 1. The initial model, termed the Baseline model (1), serves as a benchmark to evaluate the predictive efficacy of models informed by theoretical considerations[17]. It focuses exclusively on household-level characteristics and district-level structural factors, which are crucial for understanding the capabilities and desires to migrate. The model includes different measurements for five distinct livelihood assets: physical, financial, natural, social, and human capital, all of which are listed in Table S6 and theoretically motivated in the Model section of the Methods. Note that all household-level characteristics are imputed to fill in missing values between survey years. Figure S2 presents the correlation matrix of these variables, indicating no evidence of multicollinearity.

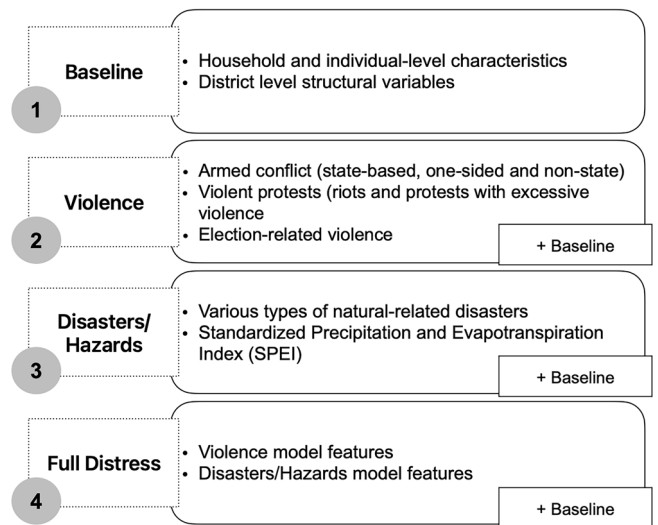

**Fig. 1 | Model specifications.** The figure summarizes the four model setups used in the analysis: baseline, violence, disasters/hazards, and full distress. Each box lists the variables included at each stage, with the "+ Baseline" labels indicating that all baseline features are retained in subsequent models.

### Table 1 | Model performance across specifications

| Model | AUROC | AP |
| --- | --- | --- |
| Baseline | 0.724 | 0.110 |
| Violence | 0.749 | 0.134 |
| Disasters/Hazards | 0.744 | 0.128 |
| Full Distress | 0.759 | 0.146 |

Average AUROC (Area Under the Receiver Operating Curve) and AP (Average Precision) scores across years, households, and 250 repetitions for the Baseline, Violence, Disasters/Hazards, and Full Distress models.

In addition to the Baseline, we further specify three distress models to examine how political and climate-related insecurities affect migration decisions. Given the presence of endogenous processes, isolating causal effects is inherently challenging. Rather than attempting to infer causality, we thus prioritize predictive modeling, which allows us to capture complex, interdependent relationships and improve generalizability[16]. However, since each distress model also incorporates baseline model features, it is possible that the predictive influence of violence and climate events on mobility, mediated through household-level variables, is attenuated, potentially leading to a more conservative estimate of their overall contribution.

The Violence model (2) integrates indicators of violent incidents, ranging from riots and excessively violent protests to election violence and armed conflicts, all listed in Table S7. The Disasters/Hazards model (3) accounts for various forms of natural hazards and climate-related disasters, including flood, drought, landslides, and storms (Table S8). Figures S3 and S4 present the correlation matrix of these variables, indicating no evidence of multicollinearity. We measure all conflict types and natural hazards as decay functions, which quantifies the diminishing impact of an event over time, where the effect of an event is halved every 6 months. To match the unit of the analysis, the variables are lagged by 1 year after aggregating them to the household level. We believe a decay function better captures the short- and long-term impacts of past events, providing a more nuanced understanding than a binary measure and reflecting real-world adaptations such as improved risk reduction and early warning systems. Additionally, descriptive statistics provided in Tables S9 and S10 detail the types of violent conflicts and disasters affecting households and demonstrate sufficient variation over time. Recognizing research that highlights the compound vulnerabilities arising from the interplay between armed conflict

and climate events[6], our Full Distress model (4) combines features from both the Violence and Disaster/Hazards models. We capitalize on the capacity of machine learning algorithms to identify interactions among various features without the need for explicit specification of interaction terms.

All models are estimated based on extreme gradient boosting, relying on the XGBoost implementation to classify binary outcomes (see SI, Fig. S5 for a graphical representation of the framework)[34]. While machine learning algorithms allow for high modeling flexibility, they are often called "black box" models, and criticized for lacking transparency and interpretability. This has led to a diverse set of model-agnostic interpretation methods[35]. The SHAP (SHapley Additive exPlanations) framework provides one such avenue[36].

SHAP, a post hoc explanation technique, draws from cooperative game theory, treating feature values as "players" and individual predictions (minus the average prediction) as "payouts"[35]. SHAP values determine each feature's importance in generating the "payout" by evaluating all possible feature combinations. Specifically, they are derived from a weighted average of the marginal contribution of each feature across all possible combinations of features[36]. Due to the vast number of potential combinations in machine learning models, we use an approach that approximates SHAP values using sample coalitions (see Supplementary Methods, Section "Estimation and Evaluation Strategy", for details)[35]. For capturing feature interactions, we employ the TreeSHAP algorithm, which assesses the joint contribution of paired features to the prediction of migration, alongside the primary effect of each feature[37].

### Predictive performance

A clear evaluation strategy becomes essential in research designs inspired by machine learning, as opposed to traditional null hypothesis significance testing[38]. Our analysis unfolds in two primary sections: first, assessing the predictive performance of each model, and then applying methods for machine learning interpretability.

We assess the models' predictive accuracy by comparing their averaged area under the receiver operating characteristic (AUROC) and average precision (AP) scores. Additional performance metrics and performance across divisions are reported in Table S12 and Fig. S6. Given the challenges in predicting at the household level and the fact that migration is a rare event in the data, we prioritize understanding the relative performance of the models, i.e., comparing the performance between the models, over their absolute predictive accuracy. Absolute predictive performance is in line with other efforts using household-level data in rare-event settings[39], and above random expectations. In this context, we use SHAP values not as causal estimates, but as a transparent way to show how the model relies on different features in its predictions, highlighting which variables matter most, whether they increase or decrease predicted migration, and whether these patterns remain stable across multiple repetitions. This approach acknowledges the inherent trade-off between model performance and interpretability; highly accurate models may be less transparent, complicating theoretical interpretation[35]. To address the uncertainty in the relative improvement among different model specifications, we calculate 95% confidence intervals by analyzing the score differences across 250 iterations.

Table 1 indicates that in order to more accurately predict household migration decisions, models must consider both the form of conflict and natural hazards to which the household has been exposed. This is seen by the Full Distress model outperforming the Baseline and the thematic Violence and Disasters/Hazards models, achieving an AUROC score of 0.76 and an AP score of 0.146.

Examining model performance differences, particularly within 95% confidence intervals, reveals valuable insights into their relative strengths (see Fig. 2). The Full Distress model shows the largest improvement, with AUROC and AP increases of 0.035 and 0.036, respectively. While these represent absolute gains of roughly 1–2 percentage points, they correspond to a relative improvement of about 9–14% in correctly identifying positive cases across thresholds compared to the Baseline. Notably, Full Distress also

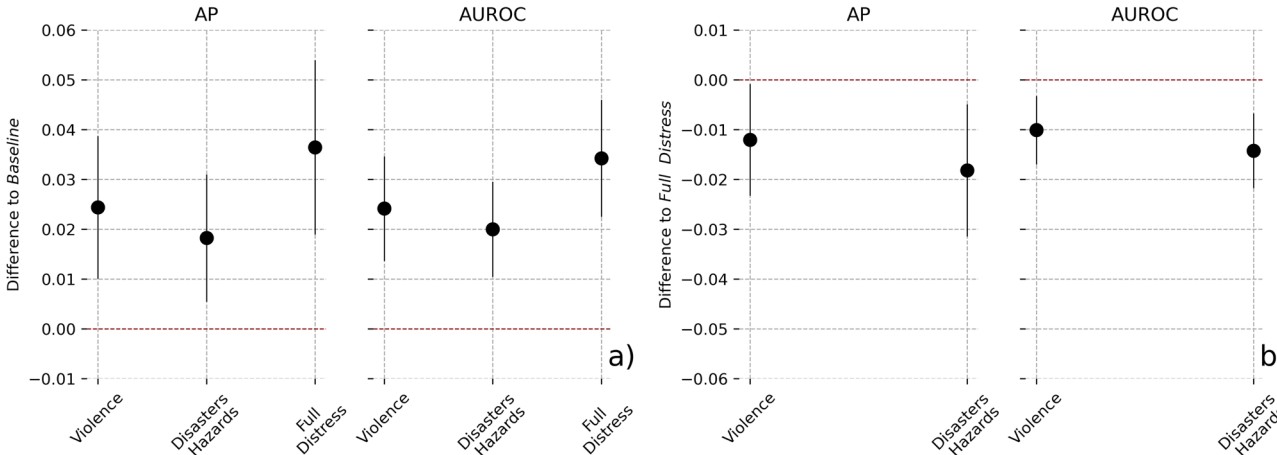

**Fig. 2 | Predictive performance improvements across models.** Changes in average precision (AP) and area under the receiver operating curve (AUROC) scores compared to the Baseline model for Violence, Disasters/Hazards, and Full Distress models (**a**), and against the Full Distress model for Violence and Disasters/Hazards

(**b**). Shaded areas indicate 95 percent confidence intervals from 250 training and testing repetitions. In (**b**), negative values signify performance improvements of the Full Distress model relative to the comparison models.

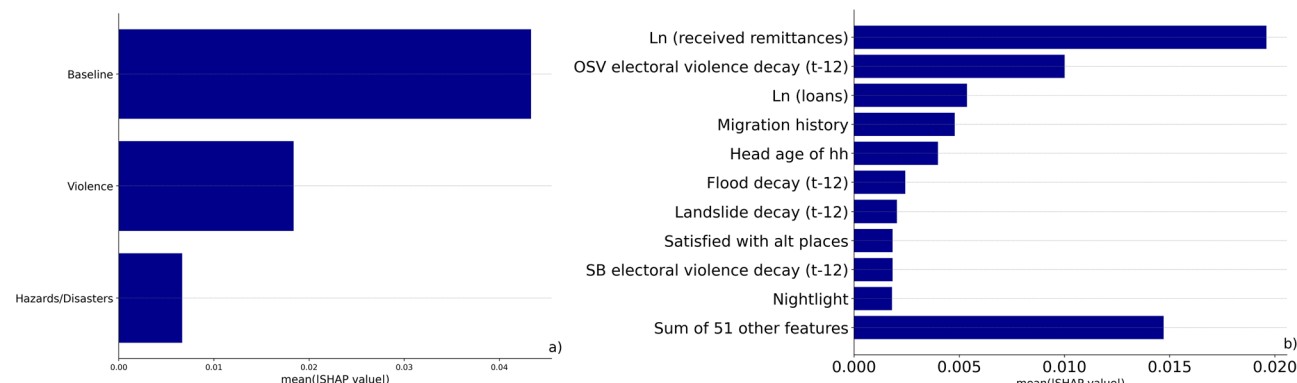

**Fig. 3 | Feature importance in the Full Distress model.** Mean absolute SHAP (SHapley Additive exPlanations) values, expressed as probabilities, for the Full Distress model: (**a**) summed by feature group and (**b**) for the ten most important features. The *x*-axis represents the average impact of each feature on the model's

output, calculated across all dataset observations and averaged over 100 repetitions. The *y*-axis lists features in order of decreasing importance. The "Sum of features" category aggregates the SHAP values of all remaining features in each model. Features related to violence and natural hazards are lagged by 12 months (t-12).

outperforms thematic models, with conflict and natural hazards together yielding AUROC and AP increases of 0.01–0.015 and 0.012–0.018. Overall, these findings demonstrate that incorporating violent events and natural hazards, individually and especially together, enhances our ability to predict household-level migration.

## Influential predictors

In addition to noting an overall increase in predictive accuracy from including violence and natural hazards separately and together, it is vital to understand how these features influence predictions of household migration decisions. As illustrated in Fig. 3, initial insights come from identifying key predictors via mean absolute SHAP values, capturing how much each feature changes the predicted absolute migration probability on average. In comparison to other feature-importance methods, SHAP values focus on the magnitude of feature attributes rather than the decrease in model performance[35]. Given that SHAP values may differ across models trained on separate data partitions, the figure displays these absolute mean values, averaged across 100 repetitions. See Figs. S14 and S15 for a comparison of results from the logistic regression models.

When considering the total effect of violence and disasters in addition to the baseline features by summing each feature's individual mean absolute SHAP value, we observe an overall larger impact of violence on the

predictions. Violence changes the predicted absolute migration probability on average by 2 percentage points (see Fig. 3a). One-sided electoral violence and floods emerge as the most significant predictors after baseline features in the thematic models and retain their influence in the Full Distress model (see Fig. 3b). Landslides and state-based election violence are also among the top 10 predictors in this model. It is important to note that both migration and distress events, such as conflict and disasters are relatively rare, which makes the consistent emergence of these predictors especially noteworthy. This rarity also implies that even modest average changes in predicted migration probability may reflect meaningful underlying patterns. However, Fig. 3, while showing the total impact on migration, does not indicate the direction of the relationships, which we explore further down in our analysis.

We now delve into how key and theoretically significant predictors influence outcomes by combining feature importance with feature effects. Our analysis concentrates on predictors that were most impactful as identified in Fig. 3. For the results presented here, we pool SHAP values across all 100 repetitions and display a random subsample of the pooled observations (10,000 draws) for readability in the plots. For robustness, we also present the results from pooling and sampling the SHAP values across repetitions with above-median out-of-sample accuracy (AP score) to ensure that the observed patterns are not diluted by poorly performing folds (see Fig. S8). Figure 4 plots the Shapley values per feature and individual observation for

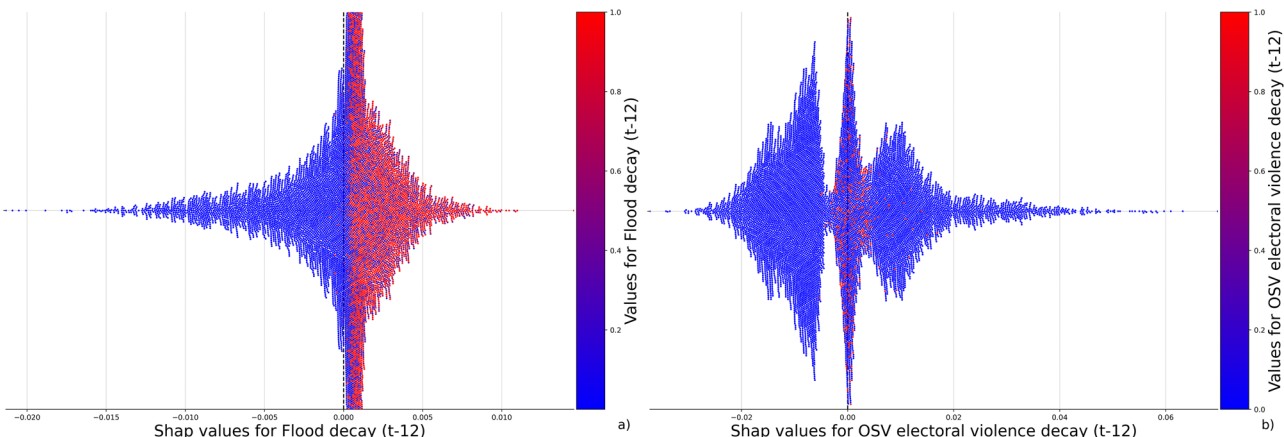

**Fig. 4 | Shapley value distributions for key predictors.** The x-axis shows individual Shapley values, expressed as probabilities, from the Full Distress model for (**a**) time since flooding decay and (**b**) time since one-sided electoral violence decay, each lagged by 12 months (t-12). Each dot represents an observation. The *y*-axis displays the distribution of data points across Shapley values, indicating how these features influence model predictions. Dot colors represent actual variable values, with red denoting high values and blue denoting low values (as shown in the color bar).

two specific variables. The impact of each observation on the model's average predicted probability is plotted on the x-axis, with the feature's actual value denoted by the color.

Figure 4a reveals that households affected by recent flooding show a higher predicted probability of migrating. The decay function for flooding indicates the elapsed time since the last flood event: shorter intervals result in higher values. Thus, households with more recent flood experiences (indicated by red) appear on the right side of the x-axis in Fig. 4a. Floods, as rapid-onset disasters, prompt significant behavioral shifts due to being extreme events. They pose risks of fatalities, as well as widespread damage to property and infrastructure, directly impacting individuals' well-being. In Bangladesh, families might first attempt to adapt to slow-onset events like droughts before contemplating migration, given the financial and emotional costs associated with relocating. However, the immediate and severe effects of flooding on well-being and the limits of household adaptation strategies often make migration a more considered response.

In addition, recent incidents of election-related one-sided violence are observed to increase households' predicted probability of migrating. The decay function for violence quantifies the time elapsed since the last incident: shorter intervals signify higher values (red color). This correlation is depicted in Fig. 4b, where households affected by more recent violence (marked in red) are mainly positioned on the right (positive) side of the x-axis, indicating a propensity to migrate. Election-related violence targeting civilians disrupts daily life, heightens fear, and signals broader political instability, potentially driving households to seek safety and stability elsewhere. Economic disruptions and social fragmentation further exacerbate insecurity, as limited job opportunities and deepened community divisions make staying less viable. Together, these factors may push households to migrate in search of security and economic stability amid escalating tensions in line with existing research that underscores the role of violence in driving migration[40–42].

**Heterogeneity and interactions**

We now shift our focus to understanding how recent exposure to violence and disasters contributes to predicting household mobility, while also considering key household-level characteristics that shape the model predictions. We particularly examine the role of one-sided violence, floods and landslides given their substantial predictive relevance (see Fig. 3b) and highlight three critical household attributes that emerge as influential predictors in our feature importance analysis: remittances, loans, and migration history of the household.

Figure 5 presents the feature importance for these household attributes, split by whether households were exposed to violence or disasters in the previous year. The x-axis represents the mean SHAP values, expressed as probabilities, for each predictor per cohort, averaged across all repetitions. See Fig. S10 for results based on averaging only over the repetitions with above-median out-of-sample accuracy. Notably, the overall magnitude of the feature impacts is smaller compared to the mean absolute impact of each feature (see Fig. 3b). This is due to the accounting for the directionality of effects, which highlights the heterogeneity within these cohorts. In other words, positive and negative effects are combined, meaning that when a feature contributes both positively and negatively across households, its overall net impact appears smaller in magnitude.

Figure 5a shows that in the context of one-sided violence, remittances are generally associated with a higher predicted probability of migration across households. We note, however, a larger predictive impact for households recently exposed to violence compared to those unexposed in the previous year. This pattern suggests that remittances play a key role in shaping migration predictions during periods of one-sided violence, potentially reflecting both financial resources for mobility and the presence of supportive social networks in the destination areas. For those unexposed in the previous year, the smaller impact highlights higher variability in the direction of the impact on the predicted probabilities, suggesting that remittances may also enhance local livelihoods. A similar pattern is observed in Fig. 5c for recent exposure to landslides, though with a larger predictive impact, indicating that remittances may be particularly critical for migration following severe natural hazard-related disasters, where local recovery is less feasible. By contrast, Fig. 5b shows a smaller and more variable impact of remittances on predicted migration probability in the context of recent flood exposure. This may stem from floods' typically temporary and localized nature, allowing households to use remittances for local recovery rather than migration. Additionally, many households can adapt or temporarily relocate nearby, making migration responses less consistent.

Regarding loans, Fig. 5 indicates a nuanced impact on the predicted probability of migration for households recently exposed to violence or disasters. For these households, loans generally show a reduced likelihood of migration, particularly in the context of one-sided violence (a) and landslides (c). This suggests that, during crises, the financial burden of loan repayment may in particular inhibit mobility by constraining households financially, reducing their flexibility to pursue migration as a viable response.

As for migration history of the household, Fig. 5 reveals a relatively small and more inconsistent predictive contribution across all events and cross-validation repetitions when households are split into the two cohorts. While the estimates generally point toward a higher predicted migration probability for households with prior migration experience, the difference

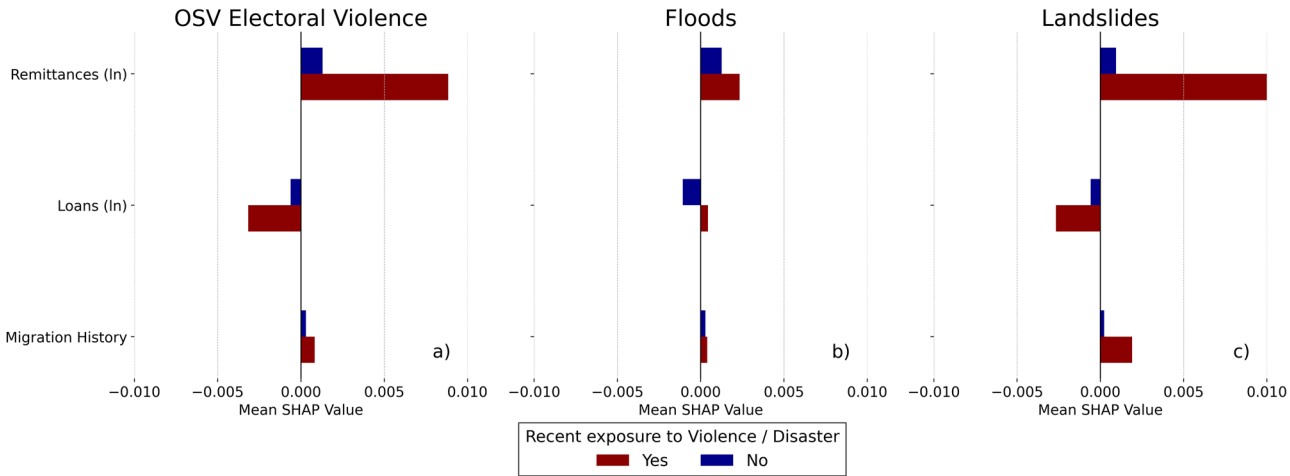

**Fig. 5 | Analysis of exposure effects on household mobility.** Mean SHAP (SHapley Additive exPlanations) values, expressed as probabilities, from the Full Distress model for cohorts recently exposed to one-sided electoral violence (**a**), floods (**b**), and landslides (**c**). The *x*-axis shows the mean net SHAP value for each feature, including remittances, loans, and migration history, among households with recent exposure (shown in red) and without exposure (shown in blue), averaged across 100 repetitions.

**Fig. 6 | Interaction effects between violence, hazard, and household features.** Sum of mean absolute SHAP (SHapley Additive exPlanations) interaction values, expressed in log-odds, between feature groups representing violence, hazards/disasters, and household characteristics, as analyzed in the Full Distress model. The *y*-axis shows the average impact of these interactions across the dataset, averaged over 100 repetitions.

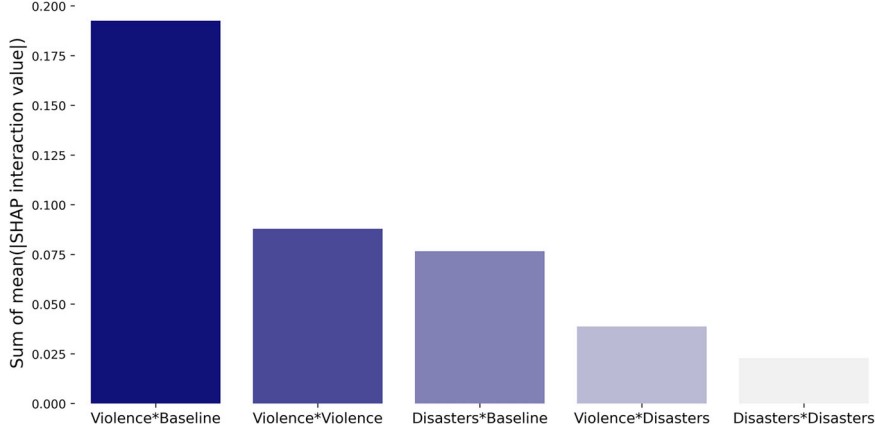

across cohorts is most pronounced in the case of landslides, suggesting, similar to remittances, the importance of supportive networks in destination areas. The patterns for exposure to violence and floods are, however, less consistent.

Subsequently, we utilize the machine learning models' ability to discern complex relationships, especially by capturing interactions between different feature groups. Figure 6 illustrates the sum of the mean absolute SHAP interaction values, averaged across all repetitions, quantifying how much different feature groups, including household characteristics, violence, and disasters, influence each other in shaping the model's predictions. See Fig. S13 for results based on averaging only over the repetitions with above-median out-of-sample accuracy. More specifically, SHAP interaction values are based on Shapley values, but instead of attributing a feature's contribution to a prediction, they decompose the contribution into independent and interaction effects, thus capturing how two variables jointly impact the prediction outcome, in addition to having independent effects. Summing these interaction values across all features provides a global measure of their cumulative impact, which is particularly relevant when analyzing models with many predictors that individually have small but collectively meaningful effects on the outcome.

The interaction between violence and household characteristics (first column in dark blue) stands out as the most influential interaction, further supporting previous findings that the predictive effect of violence on migration is not uniform across all households but is instead conditioned by specific household-level factors. The machine learning algorithm also picks up interaction effects between different types of violence, as well as between disasters and household characteristics, although to a smaller extent. Interactions between violence and disasters show smaller but meaningful effects, indicating that these factors jointly shape migration dynamics rather than acting independently. This further supports the finding that combining both improves model performance by capturing a broader range of mobility drivers.

## Discussion

In this article, we examine how natural hazards and different forms of violent conflict influence household migration patterns, both separately and in combination. Guided by a conceptual framework of aspiration and ability, we apply a predictive modeling approach to household-level data. While prior research has studied the effects of climate and conflict on migration, the interactions between these factors remain underexplored. Rather than estimating causal effects or forecasting migration counts, we assess how compound risks shape migration probabilities across different household types.

Our main findings can be summarized in three key points. First, comparing various thematic models reveals that a model incorporating both different forms of violent conflict and natural hazards outperforms simpler models. This suggests that accounting for both violent events and natural hazards enhances our ability to accurately predict migration likelihood at

the household level. Given the relative rarity of both migration and distress events at the monthly level, the predictive strength of these features shows their substantive influence on household decision-making. While this result may seem intuitive, it has not, to our knowledge, been systematically tested in previous studies.

Second, while political violence or natural hazards shape migration predictions, exposure to these risks alone does not fully determine whether individuals leave their homes. Substantial variation exists in the predicted mobility patterns among households exposed to the same event, such as one-sided electoral violence or floods. Our analysis reveals that households receiving high remittances and experiencing one-sided violence or rapid-onset events, like landslides, are more likely to send a member away, suggesting that financial resources play a key role in enabling mobility. Within the aspiration-ability framework, remittances can be interpreted in two ways: they may ease financial constraints and thus expand a household's ability to migrate, but they might also shape aspirations by signaling the success or feasibility of migration, especially if they originate from former household members who have already relocated. In this way, remittances could simultaneously enhance both the desire and the capacity to move, even if we cannot fully disentangle these mechanisms with the current data. Conversely, we find that households burdened by loans in the aftermath of one-sided violence or landslides exhibit lower predicted probabilities of migration, possibly reflecting financial constraints due to loan repayment that limit mobility options. Here, limited ability may suppress migration even in cases where the aspiration to move is present. Our analysis also incorporated variables reflecting non-material factors, including emotional bonds to community, satisfaction with life, and engagement in local activities to gauge attachment to home. However, our analysis did not reveal strong predictive patterns for these variables. Within the aspiration-ability framework, this suggests that our proxies for aspiration and place attachment may not be sufficiently precise to capture the complex ways in which emotional and motivational factors influence mobility decisions. It is also possible that, in the face of severe shocks, such considerations are outweighed by more immediate economic or safety concerns.

Finally, our study also uncovers important interaction effects that influence migration predictions. The most significant interactions occur between violence and household characteristics, followed by different forms of violence, disasters and household characteristics, violence and disasters, and finally, different types of disasters. Although interactions between violence and natural hazards have a smaller impact on predictions, they reinforce the original finding that predictability improves when both types of insecurity are considered. In particular, in Bangladesh, the combination of one-sided violence and landslides tends to amplify migration likelihood. This overlap as a dual threat of insecurity may strain community resilience and deplete resources, leaving affected households with fewer options for recovery and increasing the need to migrate. Detecting these interactions, despite the challenges posed by rare events and numerous features in our model, is essential for refining our understanding of migration drivers.

Focusing on the case of Bangladesh, the analytical framework has permitted us to examine different predictors of human mobility and their interactive nature. Violence has long been part of the political landscape of Bangladesh. This persistent sense of insecurity induced by political hostility is a reality found in many other countries, and we expect such politicized landscapes and related violence to continue to play an important role in mobility decisions. In addition, globally, but especially in Bangladesh, climate change is expected to increase environmental events such as flooding, intensify heat stress through rising temperatures, shift rainfall patterns, and cause sea level rise[43]. Predicting future mobility patterns in the face of climatic stressors is complex. Relationships derived from past data may not hold in the future, influenced by changes in economic structures, governance types, and technological advancements. Nonetheless, understanding how various crises interact can enhance our current models of predicting mobility. The Intergovernmental Panel on Climate Change report (2022) stated that the projected increase in the magnitude of extreme weather events will contribute to mobility[43]. Yet, existing models often overlook the

potential for political instability or violent conflict to compound the effects of climate change, introducing unforeseen challenges to the system[44]. While our study relies on detailed household survey data, the approach can be adapted to other countries using publicly available datasets, especially at aggregated spatial scales. Even in data-scarce settings, census data and district-level socio-economic indicators can provide a useful basis for applying similar analytical frameworks. Assessing the different trajectories of climate change, political stability, and human mobility in decades from now is quite challenging, but also a pressing step for the future research agenda. Exploring future scenarios, incorporating conflict risk[45] and political decision-making[46] offers a promising approach to address this complexity and is an essential direction for upcoming studies.

Our study contributes to the broader discourse on distress-related mobility amid global climate change and political instability. Ending distress migration requires effective disaster risk reduction and an end to political violence. Yet, given the difficulty of fully preventing such crises, developing alternative adaptation strategies remains essential. This research is an initial step in examining the combined impacts of natural hazards and violent conflict on migration, highlighting the urgent need for further studies to unpack these complex dynamics across varied household characteristics and to explore the underlying theoretical mechanisms at play.

## Methods
### The case of Bangladesh
For some countries, the stakes of calamities occurring at the same time are particularly high. Bangladesh exemplifies this, being densely populated and low-lying, with a large portion of its economy and livelihoods reliant on agriculture[47]. Freshwater flooding during the monsoon season in the delta region, while beneficial for soil fertility and agricultural output, contrasts with the challenges faced in other parts of the country, such as floods, cyclones, droughts, and shoreline erosion[43]. The government of Bangladesh has projected that "the greatest single impact of climate change might be on human migration/displacement", affecting one in seven individuals in Bangladesh by 2050 p. 4[48]. Bangladesh's Perspective Plan 2021–2041 in fact factors in climate change as a driver of future migration, while also recognizing migration as a potential adaptation mechanism[49]. Despite the implementation of various adaptation and disaster risk reduction policies, most strategies remain reactive rather than proactive. Consequently, despite a long history of established mobility patterns, there are instances where individuals are compelled to move unexpectedly due to sudden disasters.

Environmental challenges have reshaped the landscape for livelihoods and migration decisions. In Bangladesh, migration is a well-established practice, with international migration seeing a significant raise since the country's independence in 1971. Remittances from these migrants play a crucial role in bolstering the national economy. Concurrently, internal movements, both rural-to-rural and rural-to-urban, including those driven by distress, have deep roots in the country's history[49]. This indicates that, amidst natural hazards and political instability, people move along long-standing patterns of mobility.

Beyond climate variability, Bangladesh faces significant political insecurity, shaped by bipartisan rivalry between the Bangladesh Awami League and the Bangladesh Nationalist Party[27]. This competition has fostered a politically hostile atmosphere, with violence becoming an institutionalized aspect of the political culture[50]. Both parties have engaged in tactics like hartals (strikes), parliamentary boycotts, and political violence to undermine each other and secure electoral victories[51]. Such a politicized landscape and related violence between supporters of both parties have decreased the sense of security and authority at all levels. Since the elections of 2014, Bangladesh has also witnessed a weak democratic order, entering the domain of authoritarian rule, as competitive elections have been undermined and political participation has declined significantly[27]. In recent years, the Bangladeshi government has also launched an anti-terrorism operation against the Islamic State, further escalating the ongoing armed conflict between these two actors[52].

## Limitations of our mobility data

Utilizing household-level data offers both practical and theoretical benefits. Practically, surveys at this level typically experience lower attrition rates compared to individual-based surveys, since a household remains in the sample even if a member migrates. Originally surveying 6500 households in the first wave, the dataset reflects a 1.26% attrition rate by the last wave, retaining 5503 households. Table S3 in the SI displays t-test results indicating statistically significant differences on a few variables between households that dropped out and those that remained in the sample. Given the focus on the dual impact of different forms of violent conflict and natural hazards on human mobility, some attrition may reflect households that have migrated entirely or newly formed households, both of which are consistent with the dynamics under study. While this may indicate some underreporting of migration events, the low attrition rate (1.26%) suggests that any resulting bias should not significantly affect the validity of our results. Our study focuses on households consistently present in all waves.

Theoretically, research indicates that migration often occurs in stages —either the household head moves first to secure a stable situation at the destination before the rest of the family joins, or younger members are sent ahead. This pattern is evident in both regular and distress-induced mobility due to violence[19]. The new economics of migration literature further suggests that migration decisions are made collectively within the family context, not by individuals in isolation[53]. Thus, our analysis contrasts non-migrant households with migrant-sending households, acknowledging the household as the primary decision-making unit. However, this approach does not allow us to fully address intra-household disparities, such as those based on gender, due to the limitations of our data structure.

Further, our survey data captures a wide range of migration behaviors, including long-term labor migration, educational migration, and family reunification, not just distress mobility. However, distress migration often occurs within broader migration trends. This broader data provides essential context for understanding how distress migration fits into the overall migration landscape and the interplay of economic, environmental, and social factors driving it.

## Estimation strategy

XGBoost, a decision-tree-based approach for predictive modeling, iteratively learns from the training data by segmenting subsets and fitting decision trees with varying variable sets evaluated at each node. This process, where each decision tree's prediction contributes to the final output, exemplifies an ensemble learning strategy. Unlike bagging, XGBoost employs boosting as its ensemble method, benefiting from sequentially fitting models and cumulatively adding their predictions. This iterative process minimizes classification error by ensuring each subsequent tree addresses the inaccuracies of its predecessors, adjusting weights for observations that were previously mismanaged. This method effectively reduces errors through successive corrections, enhancing the model's accuracy[34].

To mitigate overfitting and boost out-of-sample predictive accuracy, we apply hyper-parameter tuning for each model, supplemented by early stopping within a five-fold cross-validation framework. A comprehensive explanation of this methodology and the final parameter specifications are detailed in SI, 1.5: Estimation and evaluation strategy. For model training, 80% of the data is used, while the remaining 20% forms an "unseen" test dataset. This test set is generated by randomly selecting from the 5503 unique household IDs, ensuring the panel structure of the data is maintained. We repeat this sampling process multiple times, producing 250 distinct forecasts. This strategy is designed to address uncertainties arising from three key sources: the multiple imputations for missing household-level variable values, the repetitive division of data into training and testing sets, and the inherent randomness of the XGBoost algorithm itself.

## Models

The Baseline: Ability and aspiration to migrate are often affected by five distinct livelihood assets: physical, financial, natural, social, and human capital[54]. We account for these factors at the household level based on

questions covered by the BIHS. All survey items used to measure household-level characteristics are listed in SI Tables S1 and S2. One potential drawback of relying on the BIHS for the household-level factors is that we can only capture their values at the time of conducting the survey. We mitigate this by following an iterative multivariate imputation strategy, where each feature's missing values are modeled as a function of the household- and district-level variables introduced in the next paragraph[55]. Physical capital is captured by the total value of household assets, home and land ownership, loans, and savings. Given the agricultural context of Bangladesh, we also have information on the value of owned livestock and whether the household receives agriculture input subsidies. Financial capital includes the total household income and monthly expenditure. To measure social capital or social networks in the area of destination, we include the migration history of a household and capture the value of incoming remittances. Human capital includes the level of educational attainment and the type of occupation of the head of household. We further account for social identities such as ethnicity, religion and language of heads of household, as well as their age and gender. Human mobility is not only driven by maximizing economic benefit but also by many cultural, psychosocial, and emotional factors such as emotional bonds to community or place of origin, good social relations, and feelings of security[56]. To capture non-material incentives and thus people's willingness to migrate, we also include questions about general satisfaction with life and relationships in the area of origin, satisfaction with opportunities to leave for other places, as well as membership and leadership in community activities to measure attachment to home and community.

Additionally, we add district-level information, focusing on the third-level administrative divisions, by incorporating harmonized nighttime light (NTL) data as an indicator of annual economic activity[57]. Districts with higher economic development might be better prepared to respond to various types of insecurities and therefore witness less emigration. We integrate this information by aggregating the raster data with a resolution of 30-arc seconds to the district level based on polygons. We also include a measurement of the gross cell product and purchasing power parity varying across districts extracted from the cross-national data on sub-national violence (xSub)[58].

The distress model: Research indicates that exposure to violence and political insecurities, including attacks and threats, influences migration aspirations[40,59–62]. This effect is observed not only in large-scale civil conflicts but also in environments characterized by political instability and lower-intensity political violence[42]. We derive data on fatalities from state-based, one-sided, and non-state violence from the UCDP Georeferenced Event Dataset[63,64]. Additionally, we utilize the Armed Conflict Location & Event Data Project (ACLED) for information on protests met with excessive force and riots, distinguishing between violence against peaceful protesters and violence emanating from protesters[65]. Considering Bangladesh's history of political clashes, including election-related violence, we incorporate data from the Deadly Electoral Conflict Dataset (DECO), which focuses on violence linked to electoral processes, outcomes, and events with fewer than 25 battle-related deaths, based on UCDP's categorization[66]. For 2018, due to DECO data limitations, we extrapolate using 2014 data (the year of the last national elections) for district-month values. We aggregate the count of violent events and fatalities to the district-year level and apply transformations to reflect the timing of events and the intensity of violence in neighboring districts.

At the same time, research has demonstrated that having experienced environmental stress or extreme weather events also affects migratory decisions[67–69]. To capture the immediate effects of sudden disaster events caused by natural hazards, we utilize the Geocoded Disasters (GDIS) dataset, which provides spatial data on disaster locations from the Emergency Events Database (EM-DAT), including extreme temperatures, landslides, storms, and floods[70]. These disaster types are aggregated to the district-month level using the latitude and longitude coordinates of each disaster's GIS polygon centroid. We further enhance this dataset with information on flood events from the Dartmouth Flood Observatory (DFO) covering our study period[71]. More specifically, we construct a measure of

flood exposure that takes into account the time since a district has last experienced flooding. To that end, we create a decay function, calculated such that if there has been a flood event in the past, we perform the following calculation: $2^{-(m/h)}$ Here, $m$ is the number of months without flood and $h$ is the half-life parameter. This corresponds to the standard exponential decay expressed in half-life form, with the specification that the weight decreases by exactly 50% after one half-life. In our setup, the effect of a flood is halved after 6 months. We believe that using a flood decay function accounts for the immediate and lingering effects of past floods, providing a more nuanced understanding than a simple binary measure of flood occurrence. This approach also better reflects real-world conditions by showing how the impact of flooding diminishes over time, considering that areas with a long history of flooding may have developed more effective disaster risk reduction and early warning systems. To also account for variation in environmental conditions concerning gradual changes, we include data on droughts measured by the Standardized Precipitation and Evapotranspiration Index (SPEI) over a 3-month period[72]. Measuring drought over a 3-month period can smooth out short-term variability, providing a more stable and representative assessment of drought conditions. In addition, droughts often affect agriculture over extended periods, and a 3-month measurement aligns better with the seasonal cycles of crop growth and water demand, offering a more meaningful assessment of agricultural drought impacts. To identify severe droughts, we apply a cut-off of SPEI $< -1.5$ and construct a decay measure analogous to floods, defined by months since the last severe drought.

## Data availability

The data supporting the findings of this study, including all replication materials used to generate and visualize the results, are publicly available at https://github.com/maxinele/migration_decisions_bangladesh.

## Code availability

The code used to generate and visualize the results in this study, along with details of the Python packages used in the data analysis, is available in a public GitHub repository at https://github.com/maxinele/migration_decisions_bangladesh. Analyses were conducted using Python (version 3.9.23). Information on specific variables and parameters applied during data processing is provided in the repository README file.

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

## Acknowledgements

We acknowledge financial support provided by the MISTRA Geopolitics program and the European Research Council project H2020-ERC-2015-AdG 694640 (ViEWS).

## Author contributions

Both authors conceived and designed the study, set up the research framework, and interpreted the results. Maxine Leis prepared the data and performed the analysis. Kristina Petrova also contributed to data preparation. Both authors wrote, reviewed, and approved the final version of the manuscript.

## Funding

## Competing interests

The authors declare no competing interests
