## [Transparent Peer Review file · Communications Earth & Environment]

Combined models of violent conflict and natural hazards improve predictions of household mobility in Bangladesh

Corresponding Author: Ms Maxine Leis

Version 0:

Decision Letter:

Dear Ms Leis,

First of all, please allow me to apologise for the delay in sending a decision on your manuscript titled "Assessing the Interplay Between Natural Hazards and Political Instability on Migration Decisions in Bangladesh". It has now been seen by 3 reviewers, and we include their comments at the end of this message. Please note that reviewer 1 is new, reviewer 2 is the original reviewer 3, and reviewer 3 is the original reviewer 1. They continue find your work of interest, but some important points are raised. We are interested in the possibility of publishing your study in Communications Earth & Environment, but would like to consider your responses to these concerns and assess a revised manuscript before we make a final decision on publication.

We therefore invite you to revise and resubmit your manuscript, along with a point-by-point response that takes into account the points raised. Please highlight all changes in the manuscript text file.

In particular, for publication in Communications Earth & Environment, we request that you (1) provide methodological details, justify the use of flood decay model, report the precision of the regression model, explain and evidence application of decay factor to sudden onset events, and no other natural hazard, clarify sample composition and rare events, and (2) expand your discussion in regards to generalization and application of your approach to other countries and reflect on how the findings relate to your framework.

Please submit your point-by-point responses as a separate file, distinct from your cover letter where you can add responses to the Editors' comments that you do not want to be made available to the reviewers. Word files are preferred. We recommend that any figures, tables or graphs that are included in the response to reviewers are also included in the main article or Supplementary Information.

Please use the following link to submit your revised manuscript, point-by-point response to the referees' comments (which should be in a separate document to any cover letter), a tracked-changes version of the manuscript (as a PDF file) and the completed checklist:

Link Redacted

We hope to receive your revised paper within six weeks; please let us know if you aren't able to submit it within this time so that we can discuss how best to proceed. If we don't hear from you, and the revision process takes significantly longer, we may close your file. In this event, we will still be happy to reconsider your paper at a later date, as long as nothing similar has been accepted for publication at Communications Earth & Environment or published elsewhere in the meantime.

Please do not hesitate to contact us if you have any questions or would like to discuss these revisions further. We look forward to seeing the revised manuscript and thank you for the opportunity to review your work.

Best regards,

Martina Grecequet, PhD
Senior Editor,
Communications Earth & Environment
Consulting Editor
Communications Sustainability

EDITORIAL POLICIES AND FORMATTING

- Behavioural and social science
- Ecological, evolutionary & environmental sciences
- Life sciences

Furthermore, please align your manuscript with our format requirements, which are summarized on the following checklist:

<https://www.nature.com/documents/commsj-phys-style-formatting-checklist-article.pdf> Communications Earth & Environment formatting checklist

and also in our style and formatting guide <https://www.nature.com/documents/commsj-phys-style-formatting-guide-accept.pdf> Communications Earth & Environment formatting guide .

*** DATA: Communications Earth & Environment endorses the principles of the Enabling FAIR data project (<http://www.copdess.org/enabling-fair-data-project/>). We ask authors to make the data that support their conclusions available in permanent, publically accessible data repositories. (Please contact the editor if you are unable to make your data available).

All Communications Earth & Environment manuscripts must include a section titled "Data Availability" at the end of the Methods section or main text (if no Methods). More information on this policy, is available at <http://www.nature.com/authors/policies/data/data-availability-statements-data-citations.pdf>.

If a community resource is unavailable, data can be submitted to generalist repositories such as <https://figshare.com/> or <http://datadryad.org/> Dryad Digital Repository. Please provide a unique identifier for the data (for example a DOI or a permanent URL) in the data availability statement, if possible. If the repository does not provide identifiers, we encourage authors to supply the search terms that will return the data. For data that have been obtained from publically available sources, please provide a URL and the specific data product name in the data availability statement. Data with a DOI should be further cited in the methods reference section.

REVIEWER COMMENTS:

Reviewer #1 (Remarks to the Author):

Paper Summary

The paper approaches the problem of understanding migration by combining factors related to natural disaster and political violence. Under the hood of the aspiration-ability framework, the authors used the XGBoost model to explain migration outcomes using features related to household characteristics, natural disaster and political violence. They demonstrate marginal improvement in the precision of the model when factors related to both violence and disaster are considered, rather than when they are incorporated individually. However, more importantly, they provided useful insights about the interaction between these external drivers and household characteristics, which showcased the potential conflict between aspiration to migrate and ability to migrate.

Overall, I like the paper due to the topical importance and the fact they have chosen Bangladesh as a case study, which is a rather under-researched country, especially in the context of political violence as a driver of migration. I would like to see the paper published with some issues addressed, outlined below.

As I am reviewing this paper, I found that it has already been through a revision process. I will try not to bring up past points already addressed. However, if any of my comments are redundant with any previous reviewer, I apologize beforehand and the authors are free to ignore it unless I acknowledge the redundancy and specify otherwise.

Novelty and Findings

I do not think that considering both political and natural drivers is something that has not been done prior. For example, see the reference below.

<https://www.mdpi.com/2071-1050/14/11/6413>

They have considered both political and environmental factors. However, they focus on inter-country migration whereas this work focuses on migration both within and outside the country. In any case, I do not find much novelty in combining the factors.

That being said, I really like how the authors have used SHAP value to capture the interaction between different features. I think there may be some novelty in this finding in the context of migration. I have not seen prior works that try to capture local interaction between such factors this way in the context of migration. Also, I know for a fact that across many households in Bangladesh, the main limiting factor of migration is related to economic burden. So, the findings make sense to me. However, I also expected education/occupation to be significant, since the largest internal migration that happens in Bangladesh from rural areas is to Dhaka in search of better livelihood/occupation. I wonder if that could be the reason that the model is worse for the Dhaka region compared to other regions. If the authors have any insight regarding this, I would be interested to know.

Method:

While this has been addressed in one prior reviewer comment to some extent, I still question the predictive ability of the model. While I acknowledge the predictive difficulty due to the sheer biases present in the data, it is quite difficult to wrap my head around the usefulness of a precision of 0.146. Maybe reporting other metrics like recall or F1-score might help. The other possibility is to expand upon the result of the logistic regression model. While the authors have shown the coefficient scores of the regression model, they have not reported the precision/accuracy of the regression model. I would request the authors to report the precision of the regression model as well.

I am also confused about the rationale of the flood decay model. Why was 2 used as a base? Why not an exponential decay model (more common)? It could be helpful if the authors could provide a little more insight into the choice of this model.

Generalizability:

As Bangladesh is a rather under-represented country, I like that the authors have chosen this region for their case study. Could they comment on how likely their approach can be employed in studying migration in other countries? In case such survey data is not available or it is available with less rich information, can such a method still be employed by just looking at publicly available data (i.e., ACLED, DFO)? Moreover, would the authors happen to know the district distribution of the households presented in the survey data? From the appendix figure of mapping performance to the division level, it seems like there are a variety of districts covered in the household data, but I am not certain.

Reviewer #2 (Remarks to the Author):

This paper makes a simple but useful point: models that account for both conflict and natural hazards outperform simpler ones accounting only for one of the phenomena. It should be of interest to a wider audience. My remaining major comments after reading the revised manuscript are the following:

1. The authors apply a decay factor to sudden onset events, such as floods, but not to any other natural hazard. I find the justification for this unclear ("We believe a decay function better captures the decreasing impact of past events over time, ering a more nuanced understanding than a binary presence-or-absence measure... " without any references...) Since no references are provided and this is non standard, I would recommend that the authors show their results when all hazards are treated in the same way, i.e., as events occurring in the year they were registered. It seems rather arbitrary to assume a certain decay of the effects of floods rather than drought, for example, which are slow-onset events and could be assumed to have more persistent effects.

2. I recommend that the authors clarify their use of the terms « conflict » and political instability » in the text (i.e. go through the text for consistency). The heading is « Assessing the interplay between natural hazards and political instability on migration decisions in Bangladesh » but the main text uses the word « conflict » which can mean many things ranging from war to civil conflict to unrest – whereas political instability makes one think of regime changes. (The text on page 15 even uses "insecurities" .)

3. I also would find it relevant to remove earthquakes in a robustness check – especially since the authors relate their analysis to climate change.

I also have a few minor suggestions to clarify and improve the text yet further:

1. On page 3, the term "SHAP feature importance" is used but not defined for readers not familiar with machine learning. The term is explained on page 7, but should be explained when it is first used.

2. There is a problem with notation of the « no migration » and « migration » response on page 5.

3. Figure 2 (page 12): there are no units on the graphs of AUROC scores??

4. Table S5 lists an impressive number of different conflict measures – surely some of these are correlated (as shown in the

correlation matrix) – why are they all included in the final model nevertheless?

5. Page 15: “both individually and collectively”? Isn’t the model at a household level? Be more precise in the text, in general.

Reviewer #3 (Remarks to the Author):

I have reviewed the paper titled “Assessing the Interplay Between Natural Hazards and Political Instability on Migration Decisions in Bangladesh” for Nature Climate Change as Reviewer 1. My overall impression of the revised manuscript is positive. The authors clearly invested significant effort and care into revising the paper prior to resubmission to Communications Earth & Environment. Most of my previous concerns and comments have been addressed, and I believe the manuscript has substantially improved as a result.

That said, I would like to raise two minor points that, in my view, would further strengthen the manuscript if clarified or elaborated upon.

1. Clarification on Sample Composition and Rare Events

I remain somewhat unclear about the number of migrants and non-migrants included in the sample. On page 5 of the memo, it is stated that 30% of households experienced migration. However, Table S3 of the supplementary material indicates that only 0.3% of months involved a migration event. While I understand that these two figures are not directly comparable, I would appreciate a simple descriptive table outlining the number of migrants per survey wave. Additionally, it would be helpful to clarify what happens when an entire household migrates. Are such households excluded from subsequent survey waves? Since the authors include only those households that responded to all waves, this could result in systematic underreporting of migration events.

Furthermore, migration, conflict, and environmental events are inherently rare, particularly when analyzed at a monthly level. Although the manuscript briefly touches on this issue, I believe a more detailed discussion would be beneficial. Specifically, when interpreting the individual variables in Figure 3b and in the Discussion section, it would be valuable to explore the implications of these results given the rarity of such events.

2. Engagement with the Aspiration–Ability Framework

In my previous comments, I expressed appreciation for the use of the aspiration–ability framework, which I find highly relevant and appropriate for the topic. While the current version of the manuscript engages with this framework more than before, the discussion still remains somewhat superficial. I believe the manuscript, particularly the Discussion section, would benefit from a more in-depth reflection on how the findings relate to this framework. What do the results imply in terms of aspirations and abilities to migrate? Integrating this perspective more explicitly could also help contextualize the rarity of migration events within the broader theoretical lens.

** Visit Nature Portfolio's author and referees' website at www.nature.com/authors for information about policies, services and author benefits**

Communications Earth & Environment is committed to improving transparency in authorship. As part of our efforts in this direction, we are now requesting that all authors identified as ‘corresponding author’ create and link their Open Researcher and Contributor Identifier (ORCID) with their account on the Manuscript Tracking System prior to acceptance. ORCID helps the scientific community achieve unambiguous attribution of all scholarly contributions. You can create and link your ORCID from the home page of the Manuscript Tracking System by clicking on ‘Modify my Springer Nature account’ and following the instructions in the link below. Please also inform all co-authors that they can add their ORCIDs to their accounts and that they must do so prior to acceptance.

Version 1:

Decision Letter:

Dear Ms Leis,

Your manuscript titled "Assessing the Interplay Between Natural Hazards and Different Forms of Violent Conflicts on Migration Decisions in Bangladesh" has now been seen by our reviewers, whose comments appear below. In light of their advice we are delighted to say that we are happy, in principle, to publish a suitably revised version in Communications Earth & Environment.

We therefore invite you to revise your paper one last time to address the remaining concerns of our reviewers. At the same time we ask that you edit your manuscript to comply with our format requirements and to maximise the accessibility and therefore the impact of your work.

EDITORIAL REQUESTS:

****Please take care to match our formatting and policy requirements. We will check revised manuscript and return manuscripts that do not comply. Such requests will lead to delays. ****

SUBMISSION INFORMATION:

OPEN ACCESS:

Communications Earth & Environment is a fully open access journal. Articles are made freely accessible on publication. For further information about article processing charges, open access funding, and advice and support from Nature Portfolio, please visit <https://www.nature.com/commsenv/open-access>

Link Redacted

Best regards,

Martina Grecequet, PhD
Senior Editor,
Communications Earth & Environment
Consulting Editor,
Communications Sustainability

REVIEWERS' COMMENTS:

Reviewer #1 (Remarks to the Author):

I have gone over the rebuttal letter and the revised manuscript, and I believe my concerns have been sufficiently addressed. I thank the authors for adding the performance metrics I requested and highlighting the generalizability aspect in the discussion section. I do not have further comments.

Reviewer #2 (Remarks to the Author):

The manuscript contributes to our understanding of the interactions between violence, natural hazards, and migration in the

context of Bangladesh. While it cannot give causal estimates of the impact of the two factors on migration, it nevertheless offers a contribution to migration analyses on Bangladesh.

It is good that the authors point out in the revised version that their results cannot be interpreted causally. This could be mentioned directly in the text, though, rather than simply in footnote 8. Another improvement in the revised manuscript is to have included logistic regressions for comparison in the SI. But it would have been helpful for the comparison to clearly write out the equation that was estimated (clarifying time indices). I was of the understanding that the analysis was at an annual level, but in the replies to the reviewers, a monthly level is mentioned. Could you please clarify this? The analysis does measure migration at a household level by year (based on the household survey)? Why then mention the monthly level in the text (and give data on it in Table S3 in Appendix)? It is somewhat confusing.

It is also good that the authors now treat all hazards with a similar decay function (lines 154-156 on page 7), not only floods. But note that the answer to this comment does not address the point really: using an SPEI-3 months does not mean you're applying the same decay function. How can you say that all models include a decay measure for SPEI? The SPEI is by definition calculated over certain lengths (3 months, 6 months, 12 months, etc.) so please explain, in the SI maybe, how this aligns with the half-time for floods.

Including both remittances and migration history at the same time does not seem wise, given their high correlation. I've understood from your previous answers that the method used here gives a conservative estimate when there is collinearity in explanatory variables. Still, it could have been good to check the results in Figure 3 and 5 when excluding migration history (which should explain remittances, in fact). This point refers to the discussion page 14 (lines 298-304) for which I believe the results may be explained by the high correlation between migration history and remittances.

Apart from this, some small problems remain:

1. The reference to <https://www.mdpi.com/2071-1050/14/11/6413> is not the most relevant or most suited (Sustainability has been classified by some as a predatory journal.). It seems a pity to have included it now, when you were referencing well-known, substantial in-depth analyses previously.

2. Since there is already a literature dealing separately with the impact of either natural hazards or violence on migration, the main contribution is the improvement over such models and the discussion in the section "Predictive performance" (pages 8-9) still emphasizes an improvement of 9-14% into positive predictions, but the improvement in terms of AUROC is nevertheless only 1.3-2%. Just be clear in the text about this.

3. Consider removing the last sentence in this section (lines 209-210) since you do not study asylum flows nor climatic conditions as such (although you control for the SPEI). The sentence linking to this reference does not seem relevant in this context (the current analysis is not studying links between higher temperatures and asylum applications) and your predictions do not support previous research in that way.

Minor points:

In Table S6: explain how you deal with zeros before taking the log. Show the actual values in the descriptive statistics rather than the log values, which are not intuitive to interpret!

Discussion: "Guided by a conceptual framework" would be better on line 331 page 16. There is no theoretical model in the paper.

Review the sentences on page 17 in the discussion of place attachment and proxies for aspirations. It seems you repeat the same argument on line 365-368 and on the following lines, 369-372 (this may suggest... this may point to...)?

In the discussion, a reader misses a "Third, ..." You start by saying you make three contributions. One reads "First, ... Second, ..." but no "Third"? I guess it's the interaction effects mentioned from line 373 onwards?

The IPCC Assessment Report is not cited as the other references (line 396, page 18).

On line 411, I think you intend to say "adaptation strategies" rather than "mitigation strategies".

Also note that there is a typo "Shapeley value" instead of "Shapley value" in the text (line 234). Line 249 leaves out "violence" in "one-sided violence". And there is one line with added words that make no sense on line 299.

Reviewer #3 (Remarks to the Author):

The authors have successfully dealt with all the comments I have raised. I would like to congratulate the authors on this nice paper and I am looking forward to hopefully seeing this paper published in Communications: Earth & Environment.

** Visit Nature Portfolio's author and referees' website at www.nature.com/authors for information about policies, services and author benefits**

We would like to thank the reviewers and editor for constructive comments on our manuscript submission. We have sought to follow them to the best of our ability, as detailed below. We have acknowledged other studies examining joint effects, clarified the predictive ability and limitations of our models, updated our measurement of floods and droughts, ensured consistent terminology in our use of "violent conflict," discussed the implications of our findings given the rarity of the events we study, and addressed several smaller comments to improve clarity. We are grateful for the opportunity to revise and resubmit our work for consideration at *Communications Earth & Environment* and believe the manuscript has improved significantly as a result of the feedback. In this memo, we discuss how we have dealt with each of the comments we received. We also include the changes made in the manuscript in [red] and point to the relevant places in the manuscript where these changes have been made more specifically.

R1.1: Overall, I like the paper due to the topical importance and the fact they have chosen Bangladesh as a case study, which is a rather under-researched country, especially in the context of political violence as a driver of migration. I would like to see the paper published with some issues addressed, outlined below.

Thank you for your encouraging comment.

R1.2 I do not think that considering both political and natural drivers is something that has not been done prior. For example, see the reference below: <https://www.mdpi.com/2071-1050/14/11/6413>

Thank you for pointing out this helpful reference. We agree that this study has already examined both political and environmental drivers of migration together. We have now cited this paper in our manuscript on page 1 and clarified that our work builds on existing research, rather than claiming this combination is entirely new.

R1.3 That being said, I really like how the authors have used SHAP value to capture the interaction between different features. I think there may be some novelty in this finding in the context of migration. I have not seen prior works that try to capture local interaction between such factors this way in the context of migration. Also, I know for a fact that across many households in Bangladesh, the main limiting factor of migration is related to economic burden. So, the findings make sense to me. However, I also expected education/occupation to be significant, since the largest internal migration that happens in Bangladesh from rural areas is to Dhaka in search of better livelihood/occupation. I wonder if that could be the reason that the model is worse for the Dhaka region compared to other regions. If the authors have any insight regarding this, I would be interested to know.

Thank you for bringing up this interesting point. As the reviewer notes, education is included in the Baseline model but did not rank among its most influential predictors, suggesting that, compared to financial capital, education had a smaller average effect on migration predictions. Unfortunately, our model does not capture migration destinations, so we cannot assess whether education might have improved performance specifically for predicting moves *into* Dhaka. The poorer performance for Dhaka likely stems partly from the survey coverage: although Dhaka is covered in the survey, the BIHS sample is mainly representative at the rural level, where most migration originates from. Migration *out* of Dhaka is likely to be driven by additional signals that may vary from general patterns thus resulting in a comparatively lower performance.

R1.4 While this has been addressed in one prior reviewer comment to some extent, I still question the predictive ability of the model. While I acknowledge the predictive difficulty due to the sheer biases present in the data, it is quite difficult to wrap my head around the usefulness of a precision of 0.146. Maybe reporting other metrics like recall or F1-score might help. The other possibility is to expand upon the result of the logistic regression model. While the authors have shown the coefficient scores of the regression model, they have not reported the precision/accuracy of the regression model. I would request the authors to report the precision of the regression model as well.

Thank you for highlighting this unclarity. We agree that a precision of 0.146 might appear modest, but we also note in the main text that our prediction task at the household level using survey data is a challenging one, given the rarity of migration events (only about 4.5% prevalence of household-migration). In such imbalanced settings, average precision should be interpreted relative to this base rate: a random classifier would achieve an AP of about 0.045, whereas our Full Distress model reaches 0.146, which is more than three times higher. As recent work by Linke et al. (2022) shows, even models using rich, individual-level data in conflict-prone settings often produce relatively low precision-recall scores. For example, their best-performing models yielded PR AUC scores in the range of 0.201 to 0.323, depending on the model and conflict type. We have now also extended the footnote on page 8 in the manuscript to be more transparent about the value and potential limitations of our models.

As suggested by the reviewer, we now also report recall and F1-scores (see new Table S12 on page 16) in the Supplementary Information. At the default threshold of 0.5, models rarely predict migration events, leading to near-zero recall and positive-class F1. At a more appropriate threshold of 0.1, the Full Distress model achieves a macro F1 of 0.579 compared to 0.559 for the Baseline, representing a relative improvement of ~4%. While absolute values remain modest, this is expected in rare-event prediction tasks. Importantly, average precision (threshold-free) captures improvements more effectively, with the Full Distress model yielding a ~32% relative improvement in F1 over the Baseline when focusing on the positive class. We also report in-sample performance metrics for the logistic regression models (Table S13) as requested by the reviewer. As expected, the regression models perform less well than the machine-learning models, with AUROC around 0.68 and AP below 0.1, alongside low macro F1 scores. These values reflect the inherent difficulty of predicting rare migration events, but also underscore the relative advantage of our machine-learning approach.

We would like to highlight that the central contribution of our paper is not to deliver highly accurate household-level predictions as stated in the article, but to evaluate whether incorporating exposure to violence and natural hazards provides systematic improvements over socioeconomic baselines by using models that are able to capture complex relationships. In this respect, our results are clear: despite modest absolute scores, the Full Distress model consistently outperforms both the baseline and thematic models across multiple evaluation metrics over repeated experiments. This demonstrates that even in highly challenging rare-event settings, conflict and hazard exposure add measurable predictive signal.

We then leverage SHAP analyses to examine how the model uses this signal, and we find consistent patterns in the relative importance, direction, and interaction of violence, disasters, and household characteristics. To further address the reviewer's concern regarding predictive ability, and to ensure that the observed patterns are not artifacts of a single repetition or of modeling noise, we now base our entire SHAP analyses

on averages across all cross-validation folds. We note this change explicitly on page 11. In addition, we provide robustness checks in the SI Appendix (Figures S8–11, S13), where we restrict the analyses to folds with above-median predictive performance. The advantage of using the full set of folds in the main text is that it makes use of the complete information contained in the cross-validation design and avoids arbitrary selection of a single “best” fold, furthermore ensure the generalizability of our results. At the same time, the above-median robustness check demonstrates that the observed patterns are not diluted by poorly performing folds. Together, this provides a more robust approach.

The substantive findings for the analysis of influential predictors remain largely unchanged. We note, however, some refinements in Figure 5, where the direction of effects becomes more consistent across floods and landslides. Overall, the main take-away of our results remains the same, but we now have greater confidence in the robustness of the presented findings. We thank the reviewer for prompting this improvement.

R1.5 I am also confused about the rationale of the flood decay model. Why was 2 used as a base? Why not an exponential decay model (more common)? It could be helpful if the authors could provide a little more insight into the choice of this model.

We thank the reviewer for the question. We have clarified this in a footnote on page 21. Our implementation is a standard exponential decay expressed in half-life form, where s is the time or distance separation and h is the half-life. This parameterization ensures that the weight decreases by exactly 50 % after one half-life. It is mathematically equivalent to the more common base- e formulation but allows h to be directly interpreted in the units of our data, which we found more intuitive when setting decay parameters. This approach follows that used in the VIEWS project (Hegre et al., 2019). We clarified our approach on page 23.

R1.6 As Bangladesh is a rather under-represented country, I like that the authors have chosen this region for their case study. Could they comment on how likely their approach can be employed in studying migration in other countries? In case such survey data is not available or it is available with less rich information, can such a method still be employed by just looking at publicly available data (i.e., ACLED, DFO)? Moreover, would the authors happen to know the district distribution of the households presented in the survey data? From the appendix figure of mapping performance to the division level, it seems like there are a variety of districts covered in the household data, but I am not certain.

Thank you for your positive remarks regarding our case study. Our method can indeed be applied in other countries, especially if some basic spatial data are available. Even if detailed household surveys are missing, researchers can still use public datasets such as census data to explore how conflict and environmental shocks may influence migration patterns at larger geographic scales, such as districts or regions. That said, household survey data remain valuable because they allow us to understand who is more likely to move or stay, based on individual and household characteristics. However, even without household data, similar types of information related to income or education attainment can still be found at an aggregated level and used in models at the district or regional scale. We briefly discuss this on page 16 in our *Discussion* section.

We also confirm that the BIHS sample includes households from a wide range of districts across all divisions in Bangladesh. We have now included Table S5 as part of the supplementary material on p.5-8.

R2.1. The authors apply a decay factor to sudden onset events, such as floods, but not to any other natural hazard. I find the justification for this unclear. Since no references are provided and this is non-standard, I would recommend that the authors show their results when all hazards are treated in the same way, i.e., as events occurring in the year they were registered. It seems rather arbitrary to assume a certain decay of the effects of floods rather than drought, for example, which are slow-onset events and could be assumed to have more persistent effects

We thank the reviewer for this helpful comment. We agree with the concern about consistency across hazard types, and in response, we now also apply a decay specification to droughts, using a threshold of SPEI < -1.5 to identify severe drought events. This ensures that both sudden- and slow-onset hazards are treated in a comparable way, while also capturing the persistence of drought impacts on agricultural livelihoods. We have clarified this in the manuscript (page 24). Importantly, the revised specification leaves the relative ranking and importance of drought features essentially unchanged, and our main findings remain robust.

R2.2 I recommend that the authors clarify their use of the terms « conflict » and political instability » in the text (i.e. go through the text for consistency). The heading is « Assessing the interplay between natural hazards and political instability on migration decisions in Bangladesh » but the main text uses the word « conflict » which can mean many things ranging from war to civil conflict to unrest – whereas political instability makes one think of regime changes. (The text on page 15 even uses “insecurities” .)

We thank the reviewer for this helpful suggestion. To improve clarity and consistency, we now use the phrase “different forms of violent conflicts” throughout the manuscript. This term better reflects the full scope of events included in our analysis. It includes state-based and non-state conflict, one-sided violence, electoral violence, violent protest and riots events, following the classifications used by UCDP and ACLED. We have revised the manuscript accordingly and updated our main title to: *Assessing the Interplay Between Natural Hazards and Different Forms of Violent Conflicts on Migration Decisions in Bangladesh*.

R2.3 I also would find it relevant to remove earthquakes in a robustness check – especially since the authors relate their analysis to climate change.

We thank the reviewer for this suggestion. We agree that earthquakes are not directly linked to climate change and that including them alongside climate-related hazards could blur the conceptual focus of our study. In response, we have re-estimated all models excluding earthquakes and now report this specification in the revised manuscript. All models therefore include a decay measure for SPEI (see R2.1) but no longer include earthquake events.

The updated results in Figure 5 show a somewhat different pattern for OSV electoral violence and floods, as discussed in R1.4. However, Figure 1 below demonstrates that this change is not attributable to the exclusion of earthquakes, since results remain the same when applying the original “best-fold” approach.

Figure 1: Replication of Figure 5 without including earthquakes in the model based on the performance on the “best-fold” -approach.

Additional changes can be observed in Figure 6 of the main manuscript, where excluding earthquakes leads to generally smaller interaction effects between disasters and other features. This is to be expected given that one hazard type has been removed from the model. Importantly, the overall findings and substantive patterns remain robust.

R2.4 On page 3, the term “SHAP feature importance” is used but not defined for readers not familiar with machine learning. The term is explained on page 7, but should be explained when it is first used.

We thank the reviewer for this helpful suggestion. We now clarify the term “SHAP feature importance” in a footnote on page 3, where it is first introduced.

R2.5 There is a problem with notation of the « no migration » and « migration » response on page 5.

We have also corrected the notation for the “no migration” and “migration” responses on page 5 by putting the text in italics.

R2.6 Figure 2 (page 12): there are no units on the graphs of AUROC scores?

We thank the reviewer for this observation. Figure 2a and 2b combine both AP and AUROC curve changes, which share the same unit on the y-axis. Since the axes are aligned and the units are identical, we decided not to repeat the y-axis labels for the AUROC panel to avoid redundancy.

R2.7 Table S5 lists an impressive number of different conflict measures – surely some of these are correlated (as shown in the correlation matrix) – why are they all included in the final model nevertheless?

We thank the reviewer for raising this important point. We agree that many of the conflict measures we include are correlated, as different forms of violence often co-occur in time and space. In classical regression this would raise concerns about multicollinearity and unstable coefficient estimates. In our

machine-learning framework, however, this issue is less problematic: tree-based models such as XGBoost can handle correlated predictors by splitting the predictive weight between them without biasing overall model performance. Substantively, we chose to retain the full set of conflict measures because theory and prior research suggest that different forms of violence may shape migration decisions in distinct ways. Including all of them allows the model to “decide” which forms carry predictive signal, and the SHAP analyses show that only a subset consistently emerges as important. At the same time, we acknowledge that correlations can blur the attribution of importance across individual indicators. In this sense, the SHAP values for highly correlated violence measures are best interpreted as conservative estimates of their influence: when multiple correlated variables are present, the model tends to split importance among them rather than exaggerate their effects. To mitigate interpretive uncertainty, we also present aggregated feature importance at the group level (violence, disasters, household factors), which yields more robust and substantively meaningful patterns.

R2.8 Page 15: “both individually and collectively”? Isn’t the model at a household level? Be more precise in the text, in general.

We thank the reviewer for this helpful comment. The sentence was intended to say that we examine the effects of natural hazards and violent conflict both separately and in combination. We have now revised the text to clarify by saying: *“In this article, we look into how various natural hazards and different forms of violent conflict influence household migration patterns, examining their effects both separately and in combination.”*

R3.1 I have reviewed the paper titled “Assessing the Interplay Between Natural Hazards and Political Instability on Migration Decisions in Bangladesh” for Nature Climate Change as Reviewer 1. My overall impression of the revised manuscript is positive. The authors clearly invested significant effort and care into revising the paper prior to resubmission to Communications Earth & Environment. Most of my previous concerns and comments have been addressed, and I believe the manuscript has substantially improved as a result.

Thank you for the positive feedback. We appreciate your review and are glad to hear that the revisions have improved the manuscript.

R3.2 I remain somewhat unclear about the number of migrants and non-migrants included in the sample. On page 5 of the memo, it is stated that 30% of households experienced migration. However, Table S3 of the supplementary material indicates that only 0.3% of months involved a migration event. While I understand that these two figures are not directly comparable, I would appreciate a simple descriptive table outlining the number of migrants per survey wave. Additionally, it would be helpful to clarify what happens when an entire household migrates. Are such households excluded from subsequent survey waves? Since the authors include only those households that responded to all waves, this could result in systematic underreporting of migration events.

We agree with the reviewer that adding an additional table showing the number of migration events per survey wave provides useful clarity about our outcome. We therefore included a new table in the supplementary material (Table S4, on page 5) that reports the number and percentage of households (HH)

reporting migration events by year. This table shows that in any given year between 0.8% and 8.5% of households reported a migration event, while in total about 30% of households reported at least one migration at some point during the survey period. We hope that this helps clarify to the reviewer why the household-level figure in the main text (30% ever migrated) differs from the monthly frequency reported in Table S3. We also added an additional sentence in the main manuscript text on page 5, clarifying the prevalence of migration in our dataset.

Thank you also for the additional helpful observation regarding what happens when an entire household migrates. We would like to clarify that this point is already addressed in the manuscript. Specifically, we state: “*Originally surveying 6,500 households in the first wave, the dataset reflects a 1.26% attrition rate by the last wave, retaining 5,503 households.*” In a corresponding footnote, we note that some of this attrition may reflect households that have migrated entirely or newly formed households, patterns that align with the dynamics under study and could indeed contribute to underreporting. However, due to the small attrition rate (1.26%), we do not expect this to significantly affect the validity of our results. We rephrased our footnote to clarify this point.

R3.3 Furthermore, migration, conflict, and environmental events are inherently rare, particularly when analyzed at a monthly level. Although the manuscript briefly touches on this issue, I believe a more detailed discussion would be beneficial. Specifically, when interpreting the individual variables in Figure 3b and in the Discussion section, it would be valuable to explore the implications of these results given the rarity of such events.

We thank the reviewer for this important point. In the revised text, we now note that migration, conflict, and disaster events are relatively rare at the monthly level. We explain that this makes the consistent emergence of these predictors in the model particularly meaningful, and that even small changes in predicted migration probability can reflect important underlying dynamics. We discuss this when interpreting the results from Figure 3b on page 10 and again in the Discussion section.

R3.4 In my previous comments, I expressed appreciation for the use of the aspiration–ability framework, which I find highly relevant and appropriate for the topic. While the current version of the manuscript engages with this framework more than before, the discussion still remains somewhat superficial. I believe the manuscript, particularly the Discussion section, would benefit from a more in-depth reflection on how the findings relate to this framework. What do the results imply in terms of aspirations and abilities to migrate? Integrating this perspective more explicitly could also help contextualize the rarity of migration events within the broader theoretical lens.

We appreciate this suggestion and fully agree that the aspiration–ability framework should be more thoroughly incorporated into the interpretation of our findings. In the revised manuscript, we engage with this framework more directly in the *Discussion* section. We have also added a sentence in the *Introduction* (on page 3) highlighting the role of remittances and loans, in order to frontload this discussion and better prepare the reader for their use in the empirical analysis.

For example, we reflect on how remittances, associated with a higher likelihood of migration, can simultaneously mean increased ability (by easing financial constraints) and influence aspiration (by signaling the feasibility or success of migration through social or family networks). Conversely, households

with outstanding loans show a lower likelihood of migration following similar events, which likely reflects constrained ability, even in cases where the aspiration to move may exist.

We also revisit our findings on non-material factors such as place attachment, life satisfaction, and community engagement. While these variables were included to capture aspiration-related dimensions, they did not show strong predictive effects. We now discuss that this may reflect limitations in how aspiration is measured in our data, or that in high-risk contexts, emotional ties to place may be outweighed by material constraints or more urgent needs for safety and security.

References:

- Hegre, H., Allansson, M., Basedau, M., Colaresi, M., Croicu, M., Fjelde, H., Hoyles, F., Hultman, L., Höglbladh, S., Jansen, R., Mouhle, N., Muhammad, S. A., Nilsson, D., Nygård, H. M., Olafsdottir, G., Petrova, K., Randahl, D., Rød, E. G., Schneider, G., ... Vestby, J. (2019). ViEWS: A political violence early-warning system. *Journal of Peace Research*, 56(2), 155-174. <https://doi.org/10.1177/0022343319823860> (Original work published 2019)

We would like to thank the reviewers and editor for constructive comments on our manuscript submission. We are grateful for the opportunity to revise and resubmit our work for consideration at *Communications Earth & Environment* and believe the manuscript has improved as a result of the feedback. We include the changes made in the manuscript and point to the relevant places in the manuscript where these changes have been made more specifically.

Reviewer 2

R2.1 The manuscript contributes to our understanding of the interactions between violence, natural hazards, and migration in the context of Bangladesh. While it cannot give causal estimates of the impact of the two factors on migration, it nevertheless offers a contribution to migration analyses on Bangladesh.

We thank the reviewer for this acknowledgement.

R2.2. It is good that the authors point out in the revised version that their results cannot be interpreted causally. This could be mentioned directly in the text, though, rather than simply in footnote 8.

We thank the reviewer for this helpful suggestion. We have now incorporated this clarification directly into the main text, rather than only in the footnote, to make clear that our results should not be interpreted causally.

R2.3 Another improvement in the revised manuscript is to have included logistic regressions for comparison in the SI. But it would have been helpful for the comparison to clearly write out the equation that was estimated (clarifying time indices).

We thank the reviewer for this helpful suggestion. In the revised manuscript, we have included the full logistic regression specification on page 23 of the Supplementary Information (Equation 1), along with a detailed description of all terms.

R2.4. I was of the understanding that the analysis was at an annual level, but in the replies to the reviewers, a monthly level is mentioned. Could you please clarify this? The analysis does measure migration at a household level by year (based on the household survey)? Why then mention the monthly level in the text (and give data on it in Table S3 in Appendix)? It is somewhat confusing.

We thank the reviewer for this helpful clarification. The analysis is indeed conducted at the household level by year, based on the household survey data. Table S3, which presented monthly-level information, was originally included at the request of another reviewer to show the temporal distribution of migration events. To avoid confusion, we have now removed Table S3 and retained only Table S4, which reports the number and percentage of households experiencing migration events by year.

R2.4. It is also good that the authors now treat all hazards with a similar decay function (lines 154-156 on page 7), not only floods. But note that the answer to this comment does not address the point

really: using an SPEI-3 months does not mean you're applying the same decay function. How can you say that all models include a decay measure for SPEI? The SPEI is by definition calculated over certain lengths (3 months, 6 months, 12 months, etc.) so please explain, in the SI maybe, how this aligns with the half-time for floods

We thank the reviewer for this important clarification. It is correct that using SPEI-3 does not, by itself, imply the application of a decay function. Rather, we applied the same decay weighting approach to the SPEI-3 index as we did for floods and conflict events, using the decay function to model the temporal persistence of hazard exposure. In other words, while SPEI-3 captures drought conditions over a 3-month window, the decay function reflects how the influence of these conditions diminishes over time, consistent with the approach used for other hazards, as indicated in lines 581-582.

R2.5 Including both remittances and migration history at the same time does not seem wise, given their high correlation. I've understood from your previous answers that the method used here gives a conservative estimate when there is collinearity in explanatory variables. Still, it could have been good to check the results in Figure 3 and 5 when excluding migration history (which should explain remittances, in fact). This point refers to the discussion page 14 (lines 298-304) for which I believe the results may be explained by the high correlation between migration history and remittances.

We appreciate the reviewer's comment and have rerun the analyses excluding the migration history variable to assess the robustness of our findings. The updated results show that the main patterns are unchanged. Specifically, Figure 3b looks very similar to the original, with all features appearing in the same order except that migration history is omitted. Figure 5 is also nearly identical, with only migration history removed from the set of predictors.

In addition, as shown in Figure S.2, the correlation matrix does not indicate a high correlation between *migration history* and *remittances*. While related, the two variables still capture different concepts: migration history reflects prior household migration experience, whereas remittances measure ongoing financial inflows.

Given these results, we conclude that including *migration history* does not substantially affect the results or interpretations. For consistency with the rest of the analysis and variable definitions, we therefore keep the original figures in the main text.

R2.6 The reference to <https://www.mdpi.com/2071-1050/14/11/6413> is not the most relevant or most suited (Sustainability has been classified by some as a predatory journal.). It seems a pity to have included it now, when you were referencing well-known, substantial in-depth analyses previously.

Additional comment: *More than the specific journal itself, what I really have an issue with regarding the particular citation is the fact that reference (6) does not include natural hazards or violent events in the analysis, such as those used in the manuscript under review (the ACLED or UCDP databases). The only 'environmental' variable included in reference (6) is (endogenous) CO₂ emissions! Therefore, it is irrelevant to cite this article in the introduction, and inappropriate to do so in the manner which it is done on page 1. The analysis in that reference has no link to the present analysis. In this context, the authors could cite references (4) and (5) instead: "While Owain and Maslin (2018) [5] and Abel et al. (2019) [4] represent notable exceptions in examining the joint impact of political and environmental factors, such integrated approaches remain rare."*

We thank the reviewer for this clarification. We had added reference (6) in response to a previous reviewer's suggestion as an example of a study linking political and environmental factors. However, we fully agree that it does not directly address natural hazards or violent events, as analyzed in our manuscript. We have therefore removed this citation and now refer instead to Owain and Maslin (2018) and Abel et al. (2019) as relevant examples of integrated approaches.

R2.7 Since there is already a literature dealing separately with the impact of either natural hazards or violence on migration, the main contribution is the improvement over such models and the discussion in the section "Predictive performance" (pages 8-9) still emphasizes an improvement of 9-14% into positive predictions, but the improvement in terms of AUROC is nevertheless only 1.3-2%. Just be clear in the text about this

We thank the reviewer for this helpful comment. We have clarified in the revised text that while the absolute gains in AUROC are around 1-2 percentage points, these correspond to a relative improvement of approximately 9-14% in correctly identifying positive cases compared to the baseline. This clarification has been added to the "Predictive performance" section (page 7)

R2.8. Consider removing the last sentence in this section (lines 209-210) since you do not study asylum flows nor climatic conditions as such (although you control for the SPEI). The sentence linking to this reference does not seem relevant in this context (the current analysis is not studying links between higher temperatures and asylum applications) and your predictions do not support previous research in that way.

We thank the reviewer for this useful suggestion. We agree that the sentence was not directly relevant to our analysis and have therefore removed it from the revised manuscript.

R2.9. In Table S6: explain how you deal with zeros before taking the log. Show the actual values in the descriptive statistics rather than the log values, which are not intuitive to interpret!

We now also include the actual values in the descriptive statistics but have kept the logged values as they are the ones informing our prediction models (using `np.log1p()` function in Python, which ensures there are no infinite values). We also include the variables pre-imputation as we do not impute the non-logged variables in our prediction pipeline.

R2.10 Discussion: "Guided by a conceptual framework" would be better on line 331 page 16. There is no theoretical model in the paper.

We thank the reviewer for this helpful clarification. We agree that "conceptual framework" more accurately describes our approach and we have revised the sentence accordingly.

R2.11 Review the sentences on page 17 in the discussion of place attachment and proxies for aspirations. It seems you repeat the same argument on line 365-368 and on the following lines, 369-372 (this may suggest... this may point to...)?

We thank the reviewer for this helpful observation. We have revised the text to remove redundancy and merge the two overlapping points into more concise discussion (p.14, lines 379-386)

R2.12 In the discussion, a reader misses a "Third, ..." You start by saying you make three contributions. One reads "First, ... Second, ..." but no "Third"? I guess it's the interaction effects mentioned from line 373 onwards?

We thank the reviewer for this helpful observation. We have now clarified the structure by replacing the missing “Third” with “Finally,” to explicitly introduce the third contribution discussing the interaction effects (page 14, line 387)

R2.13 The IPCC Assessment Report is not cited as the other references (line 396, page 18).

We thank the reviewer for this observation. We have now included the IPCC (2022) Assessment Report as a formal citation in the text to ensure it is referenced consistently with other sources.

R2.14 On line 411, I think you intend to say "adaptation strategies" rather than "mitigation strategies".

We thank the reviewer for noticing this oversight. We agree with the suggestion and have replaced “mitigation strategies” with “adaptation strategies”.

R2.15 Also note that there is a typo "Shapeley value" instead of "Shapley value" in the text (line 234). Line 249 leaves out "violence" in "one-sided violence". And there is one line with added words that make no sense on line 299.

We thank the reviewer for carefully noting these errors. We have corrected the typo to “Shapley value,” added the missing word “violence” in “one-sided violence,” and removed the extra words on line 299.

R2.16 When comparing logistic regressions with the machine learning model in the Supplementary Information (SI), the authors only include a set of logistic regressions with violence and another set with natural hazards in Table S13. To make a full comparison, they should also include a logistic regression incorporating both violence and natural hazards, given that they argue that this model performs best among the machine learning models. This is important since, on page 16 of the SI, the authors claim that 'As expected, the XGBoost models outperform the logistic regressions, ...', based on the F1, AUROC and AP scores shown. Even if the authors find that logistic regressions with both violence and natural hazards perform better than the full machine learning analysis, the paper would still make a contribution, and it is important to show all regressions to enable a complete and transparent comparison.

We have now also included a model which includes both violence and natural hazards (see Figure S15c) and added the results to Table S13. The results show that the XGBoost analysis still outperforms the logistic regression.